# Unity is Strength? Benchmarking the Robustness of Fusion-based 3D Object Detection against Physical Sensor Attack

## ABSTRACT

As a safety-critical application, Autonomous Driving (AD) has received growing attention from security researchers. AD heavily relies on sensors for perception. However, sensors themselves are susceptible to various threats since they are exposed to the environments and vulnerable to malicious or interfering signals. To cope with situations where a sensor might malfunction, Multi Sensor Fusion (MSF) was proposed as a general strategy to enhance the robustness of perception models.

In this paper, we focus on investigating MSF security under various sensor attacks and wish to answer the following research questions: (1) *Does fusion enhance security or not?* (2) *How does the architecture of the fusion model influence robustness?* To this end, we establish a rigorous benchmark for fusion-based 3D object detection robustness. Our new benchmark features 5 types of LiDAR attacks and 6 types of camera attacks. Different from traditional benchmarks, we take the physical sensor attacks into consideration during the corruption construction. Then, we systematically investigate 7 MSF-based and 5 single-modality 3D object detection models with different fusion architectures. We will release the benchmarks and codes to facilitate future studies.

## 1 INTRODUCTION

In autonomous driving (AD), 3D object detection serves as the core basis of the perception stack, especially for the sake of path planning, motion prediction, collision avoidance, etc. LiDAR and camera are the two most important sensors for 3D object detection. LiDAR provides precise 3D spatial information through point cloud data, while cameras provide rich texture information through image data. The fusion of these two complementary sources of information is a common effort in both academia [18, 23, 24, 52, 60] and industry [1–3, 6, 7] to enhance perception performance.

However, many recent security studies [11, 12, 21, 34–36, 48, 51, 53, 56, 66, 67, 81–84] indicate that LiDAR and camera systems can be compromised by physical signals, e.g., laser, electromagnetic interference (EMI) and ultrasound. We adopt the term *physical sensor attacks* to describe attacks that employ physical signals to manipulate sensor output. The physical sensor attack will induce the point clouds and images inevitably to encounter significant corruption. As an extremely safety-critical application, autonomous driving particularly requires enhanced robustness to address the corruptions that may arise in the physical world. Many preceding works [14, 34, 36, 82] have considered sensor fusion as potential countermeasures. However, whether fusion can attenuate the attacks as anticipated remains an open question, lacking systematic research.

One common practice for such robustness analysis is to establish a benchmark [26]. Several benchmarks are proposed for image corruption [10, 31, 37] and point cloud corruption [47, 85]. As shown in Fig. 1, the corruptions used in those benchmarks can be grouped into bad weather, digital noise, sensor failure, object abnormalities,

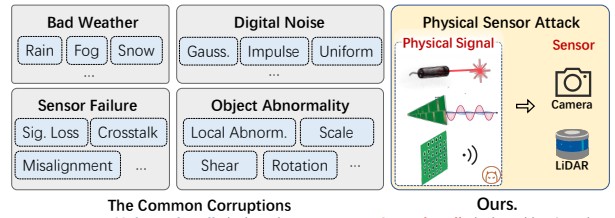

**Figure 1: The common corruptions used in previous robustness benchmark and the sensor attack-based corruptions in this work.**

etc. Compare to those corruptions, the corruptions in this work are intentionally induced by attackers with physical sensor attack. Recently, a small amount of benchmarks [20, 26] has focused on the robustness of MSF-based perception. However, none of them has considered the corruption induced by sensor attack, and the number of MSF-based 3D object detection models under evaluation is still limited, e.g., only 3 models in [26] and 2 models in [20] are tested on the corrupted dataset.

In this paper, we propose a benchmark for evaluating the robustness of MSF-based 3D object detection against 11 types of sensor attacks. Such a benchmark could provide significant value to both academia and industry. As a shared reference, it could facilitate various activities, including developer training, assessing risks, and advancing the design of new MSF-based models. Based on the benchmark, we set out to answer the following research questions:

**RQ1. Does fusion enhance security?** Compared to single-modality models, can fusion models offer enhanced security? This is a fundamental and crucial question. The field of autonomous driving, being safety-critical, is highly susceptible to being targeted by attackers. However, no study has systematically investigated the robustness of multi-sensor fusion models in autonomous driving when faced with malicious sensor attacks. This study answers the question (detaied in Sec. 4.4) by evaluating three aspects: targeted attack robustness, single source robustness, and overall robustness.

**RQ2. How does the architecture of the fusion model influence robustness?** In the face of physical sensor attacks, do multi-sensor fusion-based detectors with varying architectures demonstrate performance disparities? If such differences exist, what are the underlying reasons? Previous research typically categorizes models into early, middle, and late fusion. However, we found that this classification method does not explicitly reveal the relationship between architecture and robustness. In this paper, we introduce a novel paradigm, categorizing models based on *fusion sequence* and *fusion representation*, and delve into the relationship between architecture and robustness using the concept of information entropy. Through this new classification approach, the answers to RQ2 have become clearer (detailed in Sec. 4.5).

Building a benchmark to answer those questions is a challenging problem, especially when considering the physical feasibility and comprehensiveness of the dataset. Unlike purely digital corruptions, which allow arbitrary editing of images and point clouds,

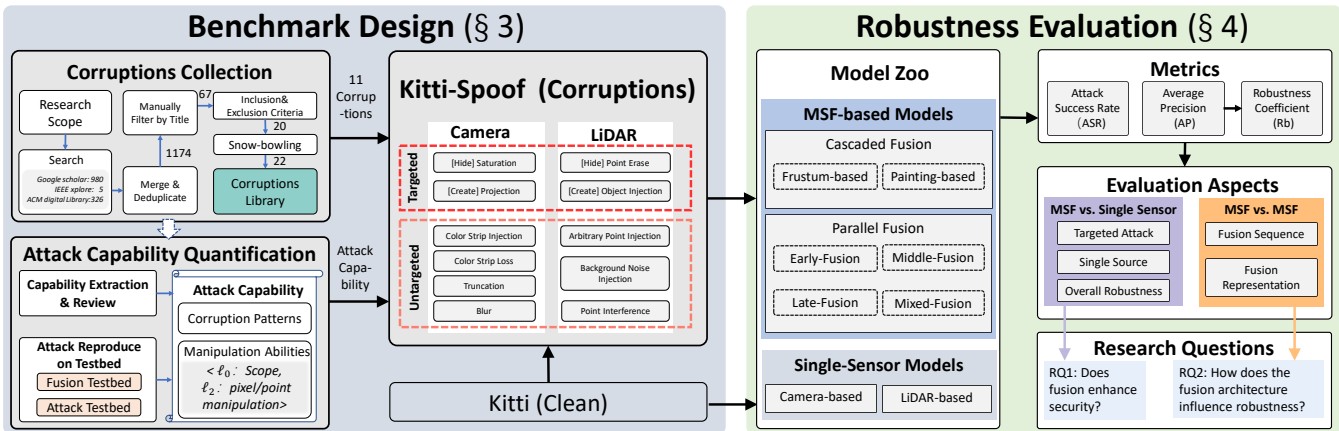

**Figure 2: The benchmark overview.** First, we collect works related to sensor attacks as comprehensively as possible through a Systematic Literature Review (SLR) process. Second, we quantify the attack capability by reviewing the paper and reproducing the attacks on our physical testbed. Third, we design sensor attack corruptions for both LiDAR and camera sensors. By applying corruptions to typical autonomous driving datasets KITTI [27], we establish the sensor attack robustness benchmark dataset KITTI-spoof. Finally, We conduct large-scale experiments centering around two research questions to benchmark the robustness of MSF-based model against physical sensor attack.

making the dataset physically realizable requires considering the capabilities of physical sensor attacks. However, previous sensor attacks have mainly focused on demonstrating their attack effectiveness but have not explicitly quantified their attack capabilities for benchmarking purposes.

To bridge this gap, we design the benchmark process as shown in Fig. 2. In summary, our contributions are concluded as follows:

- **Benchmark.** We present a large-scale robustness benchmark for MSF-based 3D object detection under physical sensor attack, namely Kitti-Spoof. The dataset contains 11 corruptions induced by laser, EMI, and acoustic.
- **Empirical Evaluation.** Based on the benchmark, we perform a large-scale (542,736 frames) empirical study to evaluate the sensor attack robustness on 7 MSF-based detectors and 5 single-modality detectors with different architectures.
- **Insights for Critical Research Question.** This paper systematically answers the fundamental and critical questions related to the robustness of MSF-based models. Additionally, we provide insights for enhancing MSF robustness.

## 2 THREAT MODEL AND DEFINITION

### 2.1 Threat Model of Physical Sensor Attack

In this benchmark, we consider adversaries with the following assumptions.

**Attack capability:** The adversary conducts attacks outside the car to be stealthy. She can aim the camera or LiDAR and inject signals to attack them.

**Sensor Assessment:** The adversary has no direct access to the target sensors. She cannot physically touch them, alter the device settings, or install malware. However, we assume that she is fully aware of the characteristics of the target sensors. Such knowledge can be obtained from the user manual or by analyzing a sensor of the same model as the target sensor.

**Black box:** The adversary does not have access to the machine learning model or the perception system. Attackers can exploit

only the characteristics and vulnerabilities of the sensors to achieve their attack target.

### 2.2 Scope and Definition of Sensor Attack Robustness

Firstly, we demarcate the scope of physical sensor attacks. Since sensors act as transducers that translate physical signals into electrical ones [25], we focus on physical signal attacks that corrupt the output of the sensor, with the threat model elaborated in Sec. 2.1. The subsequent attacks do not fall within our benchmark's scope: (1) physical modification of the measured target, such as utilizing stickers [22, 32, 40, 74, 79, 88, 92] or 3D objects [15, 71, 87] to deceive sensors, and (2) attacking the digital transmission of sensor data in CAN bus [16, 41], or sensor networks [45, 61, 80].

We now define sensor attack robustness. To begin, we consider a detector $f : X \rightarrow Y$ trained on samples from distribution $\mathcal{D}$. Most detectors are judged by their performance with the intersection of union ($IoU$) and a threshold ($t$) on test queries drawn from $\mathcal{D}$, i.e., $\mathbb{P}_{(x,y)\sim\mathcal{D}}(IoU(f(x),y) > t)$. Yet in safety-critical applications, the detector may face malicious sensor attacks and is tasked with artificially corrupted inputs. In view of this, we suggest computing the detectors's *sensor attack robustness* $\mathbb{E}_{c\sim C}[\mathbb{P}_{(x,y)\sim\mathcal{D}}(IoU(f(c),y) > t)]$, where $C$ is a set of corruptions. The design of corruptions $C$ should satisfy the physical realizability, i.e., $\|C - X\| < \Phi$, where $\Phi$ is a set of physical attack capability of sensor attacks.

## 3 BENCHMARK DESIGN

In this section, we first introduce the design methodology for the corrupted dataset KITTI-Spoof. We then detail the corruptions specific to the camera and LiDAR respectively.

### 3.1 Design Methodology for Kitti-Spoof

When designing the corrupted dataset Kitti-Spoof, we aim to ensure the comprehensiveness and physical feasibility of the dataset. The

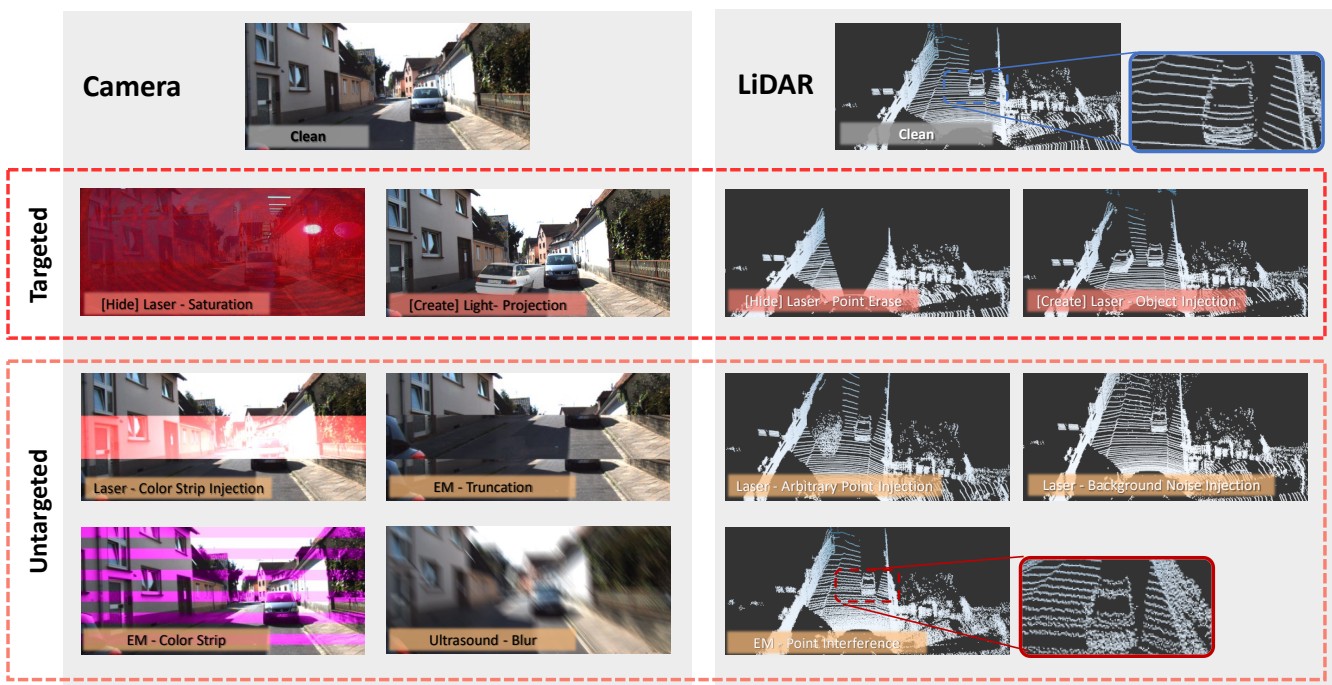

**Figure 3: The corruptions in our benchmark.** The corruptions are named in the format *[attack target (if any)] signal-patterns*. There are 6 camera corruptions and 5 LiDAR corruptions. The corruptions are grouped into *targeted* and *untargeted* according to the attack effect on single-modality detectors. Best viewed on a screen and zoomed in.

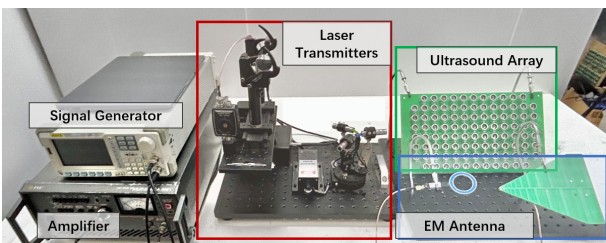

**Figure 4: Attack Testbed.**

source papers and attack capability of corruptions are listed in Table 3. Examples of corruptions are illustrated in Fig. 3.

*3.1.1 Corruption Collection.* We collected works related to sensor attacks with the scope defined in Sec. 2.2 as comprehensively as possible through a Systematic Literature Review (SLR) [39, 43] process. The SLR itself follows a three-step methodology comprising planning, conducting, and reporting, as depicted in Fig. 2. We define the search scope as "physically-realizable sensor attack" and use the following query to search for the terms in the documents.

> **Query**: ("physical" OR "real-world" OR "practical") AND "signal" AND ("attack" OR "vulnerability" ) AND ("LiDAR" OR "camera" ) AND ("autonomous driving" OR "self driving" OR "autonomous vehicle")

By leveraging the citation analysis software *Publish or Perish* [30], we collected studies from Google Scholar (980), IEEE Xplore (5),

and ACM Digital Library (326). After removing duplicates from the total of 1311 search results, 1174 papers remained.

We only included studies related to physical sensor attacks on cameras or LiDARs. Moreover, we only included papers that first introduced the attack as well as those that made improvements to the attack. Initially, we conducted a preliminary filter based on paper titles, resulting in 67 potentially relevant papers. Subsequently, after reviewing the content, we shortlisted 20 articles. We then employed the snowballing technique on all these works to uncover resources overlooked in the initial search and applied the same inclusion criteria, leading to the addition of 2 new studies.

The *SLR* process yielded a total of 22 scientific works, and we distilled 11 types of corruptions from these papers.

*3.1.2 Attack Capability Quantification.* As shown in Fig. 2, we quantify the capabilities of sensor attacks in two steps. First, we extract useful information by reviewing source papers, which can ascertain the pattern characteristics. Some papers clearly describe the capability of attacks, while others do not. Second, we replicate each attack on our physical testbed, as depicted in Fig. 4, to ensure the physical feasibility of each attack and further clarify each attack's capability and limitations.

We quantify the capabilities of sensor attacks based on the corruption pattern characteristics and manipulation abilities. Similar to adversarial attacks [17], we utilize the $\ell_0$ and $l_2$ norms to represent the manipulation abilities on images or point clouds, wherein $l_0$ norm signifies the attack scope, and $l_2$ norm illustrates

the pixel/point manipulation capability. More specifically, for images, $l_0$ denotes the number (scope) of pixels that can be manipulated by the attack, and $l_2$ indicates how can the pixel values be manipulated. For point clouds, $l_0$ represents the number (scope) of points that can be affected by the attack, and $l_2$ signifies how can the distance of the points be manipulated.

Our physical testbed consists of a sensor fusion system and a signal transmission system. The fusion testbed, as shown in Fig. 7 in Appendix, comprises a Leopard USB3.0 camera [5] and a VLP-16 LiDAR [72] mounted on an Apollo-kit. The attack testbed, as shown in Fig. 4, includes a signal generator, an amplifier, and three types of signal transmitters, which can transmit laser, ultrasound, and electromagnetic signals.

Detailed attack capability quantification process of every corruption is described in Appendix A.

## 3.2 Image Corruption

There are a total of six image corruptions in this benchmark, including two targeted corruptions and four untargeted corruptions.

**[Camera-Hide]Laser-Saturation:** The attack method [56, 83] involves using a high-power laser or a high-lumen light beam to directly irradiate the camera. This causes the light-sensitive module in the camera to be saturated, effectively hiding the real objects in the environment. The principle of this phenomenon is similar to that of overexposure in dynamic lighting conditions [49] in real life. Such overexposure is caused by excessive luminous flux and saturation of the light-sensitive module. The saturation (or overexposure) commonly occurs in real-world driving situations, such as when the car is coming out of a tunnel [70] or when an oncoming car activates its high-beam headlights [8]. In this situation, the camera's image will be overexposed and blinded.

**[Create]Light-Projection:** This attack method involves using a projector to cast images onto the environment [21, 48, 51, 53, 77] or directly projecting images into the camera [51]. While this attack method might seem somewhat naive, it represents a significant threat. Its effectiveness bears some similarity to sticker-based attacks. However, compared to such sticker attacks, it offers distinct harm. Firstly, it can be executed remotely without requiring the attacker to go over there and stick it himself. Secondly, it provides convenient control over the attack via signal manipulation. Thirdly, it can project elements into locations that are challenging for sticker-based attacks, such as trees by the roadside [53] or air [21]. Nevertheless, the primary drawback of this projection attack lies in its susceptibility to environmental lighting conditions.

**Laser - Color Strip Injection:** This attack method [65, 84] involves exploiting the rolling shutter of CMOS sensors, allowing attackers to inject a colored stripe. Prior research [84] evaluated the impact of this attack on traffic light recognition.

**EM - Strip Loss and EM - Truncation:** This attack [35] targets the camera interface bus used for image signal transmission and employs intentional electromagnetic interference (IEMI) to inject malicious signals, causing camera glitches. The principle of the attack is that cameras using MIPI CSI-2 transmission standard allocate a buffer for image signals. The start/end address of the buffer and the line pitch are passed to the Unicam (CSI Receiver). The image signals are transmitted by individual lines and decoded

based on the fixed color filter arrangement. The camera will discard the lines that encounter transmission errors. If one line in the transmission is missing, it can disrupt the color interpretation of the subsequent lines during image processing, thereby causing color strips. If the start/end address of a buffer is missing, inter-frame content stitching appears, thereby causing truncation.

**Ultrasound-Blur:** This attack [34, 89] is based on a system-level vulnerability that image stabilizer hardware is susceptible to acoustic manipulation. By emitting deliberately designed acoustic signals, an adversary can control the output of an inertial sensor, which triggers unnecessary motion compensation and results in a blurred image.

## 3.3 Point Cloud Corruption

There are a total of five point cloud corruptions in this benchmark, including two targeted corruptions and three untargeted corruptions.

**[Hide]Laser-Point Erase:** Existing research has already demonstrated the feasibility of erasing point clouds using continuous-wave laser [67]and pulsed laser [12, 36], thereby hiding targeted objects. LiDAR functions by emitting lasers and receiving echoes from objects to perform time-of-flight measurements and distance measurements, ultimately generating point clouds. Existing point erasure methods fundamentally disrupt or hide the valid echoes from objects. Shin et al. [67] utilize a high-power (800mW) continuous laser to saturate the LiDAR's photodetectors, rendering them incapable of receiving valid echoes. Jin et al. [36] and Cao et al. [12] adopt pulsed lasers of specific frequencies to inject high-intensity points and then utilize the point cloud's echo filtering mechanism to filter out valid echoes.

**[Create]Laser-Object Injection:** This type of attack [19, 36, 64] employs a set of laser receiver and transmitter for controllable point cloud injection against mechanical LiDAR systems. The PLA-LiDAR [36] proved that it's feasible to inject point clouds in the physical world and directly spoof 3D object detection models using a black-box approach.

**Laser - Arbitrary Point Injection:** Several studies [14, 29, 56, 67, 69] have successfully implemented laser-based points injection attacks against LiDAR. However, these injected points exhibit a certain level of randomness rather than regular shapes shown in papers [36] and [64]. We suppose this might be due to differences in signal design and a lack of precise synchronization compared to controllable injection. Even though these attacks have not been proven to achieve targeted attack effects in the physical world, we are curious about their potential impact on fusion model performance.

**Laser - Background Noise Injection:** This type of attack [67] involves injecting random fake points using low-power lasers. The authors demonstrate that this may be due to the low-power laser causing an increase in baseline noise. In the everyday use of LiDAR, similar noise is sometimes observed, primarily due to interference between LiDARs, which is different from the principle of random noise injection attacks.

**EM - Point Interference:** This type of attack [11] exploits the susceptibility of time-of-flight (TOF) circuits to electromagnetic (EM) waves. By injecting EM signals at specific frequencies

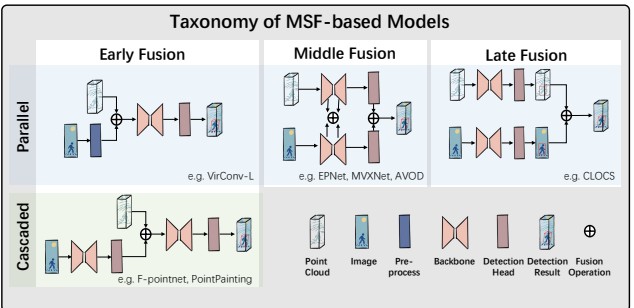

**Figure 5: MSF-based 3D object detection models with different architectures.**

into the LiDAR's circuits, it disrupts the LiDAR's ranging function, consequently corrupting the global point cloud and introducing disturbances at radial distances.

# 4 ROBUSTNESS EVALUATION

## 4.1 Fusion-based Model Collection

To collect as many appropriate SOTA MSF detectors as possible for our benchmark, we primarily focus on the MSF-related survey literature [18, 23, 24, 52, 60, 76] and collect papers published in relevant top-tier conferences and journals during the last six years. Meanwhile, we refer to the 3D object detection leaderboard of Kitti benchmark [4] for the open-source models achieving SOTA performance. Eventually, we selected 7 state-of-the-art MSF systems as shown in Table 4 in the Appendix.

The SOTA MSF detectors are primarily based on LiDAR-based 3D object detectors and try to incorporate image information into different stages of a LiDAR detection pipeline, as shown in Fig. 5. According to the different fusion stages, MSF models can be categorized into *early fusion*, *middle fusion*, and *late fusion*. There are three early-fusion models (F-Pointnet [58], Pointpainting [73] and VirConv-L [78]), two middle-fusion models (EPNet [33] and AVOD [42]), one late-fusion model (CLOCs [54]) and one mixed-fusion model (VirConv-T [78]) in our benchmark.

Most models process data from both sensors concurrently before the fusion operation. We define this concurrent approach as *parallel fusion*. In early-fusion, a type of fusion follows a sequential structure, which we refer to as *cascaded fusion*. In cascaded models, an image-based model is first used to obtain 2D recognition results, such as bounding boxes [58] or semantic classes [73]. These 2D results are then employed to enhance the point cloud, which is subsequently fed into the LiDAR-based model.

## 4.2 Metrics

In this section, we define the robustness evaluation metrics. 3D object detection aims to locate, classify, and estimate oriented bounding boxes in the 3D space. The accuracy of object detection can be measured by IoU (Intersection over Union), which measures the intersection area between a ground-truth 3D bounding box $B_g$ and a predicted 3D bounding box $B_p$ over their union area.

We consider a detection for a car successful when the IoU is larger than 0.7, which is the same as Kitti [27]. To better benchmark the

robustness of models against different attacks, we have employed several advanced metrics based on IoU.

*4.2.1 Attack Success Rate (ASR).* ASR is employed to quantify the success rate of *targeted* attacks. In our benchmark, targeted attacks can achieve two types of effects against a black-box model: hide and create. The hide attack is considered successful solely when the target object evades detection. Conversely, the create attack is deemed successful only when an initially non-existent object is generated within a designated region.

*4.2.2 Average Precision (AP).* AP approximates the shape of the Precision/Recall curve as:

$$AP\mid_{R_{40}} = \frac{1}{|R_{40}|} \sum_{r \in R_{40}} \max_{\tilde{r}:\tilde{r}>r} \rho(\tilde{r}) \tag{1}$$

where $\rho(r)$ gives the precision at recall $r$, meaning that instead of averaging over the actually observed precision values per point $r$, the maximum precision at recall value greater than or equal to $r$ is taken. We adopted mean AP ($mAP$) [63, 68], by taking the average of APs at three difficulty levels (i.e., "Easy", "Moderate", and "Hard"), to measure the overall detection performance of a model.

*4.2.3 Robustness (Rb).* We define the Robustness of one MSF model on a corruption $c$ as $Rb_c$:

$$Rb_c = \frac{mAP_c}{mAP_{clean}} \tag{2}$$

where $mAP_c$ and $mAP_{clean}$ represent the overall performance of the model on corruption $c$ and clean data, respectively. The mean robustness of one model across multiple corruptions is denoted as $mRb$.

## 4.3 Setup

We benchmark the 7 MSF-based models and 5 single-modality models using Kitti-Spoof. To ensure, as far as possible, that the models are compared on the same baseline, each model uses the official model parameters which are fine-tuned on the Kitti train dataset. Each model is tested on 12 datasets (1 clean + 11 corrupted), each dataset containing 3,769 LiDAR-camera frames. Since some of the models only support the detection of the "car" category, we calculate the performance of all models based on their detection results for cars. After obtaining the detection results for each frame, we compute the AP (shown in Table 5 in Appendix) with a 0.7 IoU threshold. Then the AP results are used to calculate the robustness $Rb$ (shown in Table 2). In addition, for targeted corruption, we calculate the attack success rates $ASRs$ (shown in Table 1). Based on these results, we engage in discussion and analysis centered around two research questions.

## 4.4 RQ1. Does fusion enhance security?

To comprehensively evaluate whether MSF-based models enhance security compared to single-modality models, we decompose security into the following three aspects:

*4.4.1 RQ1.1 ASR of Targeted Attack:* Evaluating targeted attacks is of considerable importance due to its real-world relevance. In specific driving scenarios, attackers often aim to conceal target objects or create objects at predetermined locations, potentially inducing

Table 1: Attack Success Rate

| Target Sensor | Attack Target | Corruption | Camera-only ImVoxelNet | LiDAR-only PointPillar | Fusion Model F-pointnet | Pointpainting | virconv_l | virconv_t | Epnet | AVOD | CLOCS |
|---|---|---|---|---|---|---|---|---|---|---|---|
| Camera | Hide | Saturation | 97.37% | \ | 92.58% | 47.41% | 0.04% | 0.54% | 7.78% | 20.48% | 88.51% |
| Camera | Create | Projection | 100.00% | \ | 96.77% | 0.00% | 0.00% | 0.00% | 0.00% | 0.00% | 0.00% |
| LiDAR | Hide | Point Erase | \ | 100.00% | 100.00% | 100.00% | 100.00% | 100.00% | 100.00% | 100.00% | 100.00% |
| LiDAR | Create | Point Injection | \ | 100.00% | 0.00% | 95.93% | 100.00% | 98.27% | 100.00% | 100.00% | 0.00% |

Table 2: The Robustness(Rb) of 5 single-modality Detectors and 7 MSF-based on Kitti-Spoof.

| Target Sensor | Corruption | Camera-only ImVoxelNet | SMOKE | LiDAR-only Second | PointPillar | 3DSSD | Fusion Model F-PointNet | PointPainting | VirConv_L | VirConv_T | EPNet | AVOD | CLOCs |
|---|---|---|---|---|---|---|---|---|---|---|---|---|---|
| Camera | [Hide] Laser - Saturation | 0.415 | 0.069 | / | / | / | 0.226 | 0.402 | **0.999** | 0.995 | 0.804 | 0.592 | 0.315 |
| | [Create] Light- Projection | 0.668 | 0.852 | / | / | / | 0.467 | 0.973 | 0.999 | **1.000** | 0.999 | 0.995 | 0.984 |
| | Laser - Color Strip Injection | 0.520 | 0.203 | / | / | / | 0.962 | 0.832 | **0.999** | 0.993 | 0.797 | 0.752 | 0.993 |
| | EM - Color Strip | 0.549 | 0.749 | / | / | / | 0.947 | 0.916 | 0.967 | **0.985** | 0.891 | 0.790 | 0.992 |
| | EM - Truncation | 0.010 | 0.000 | / | / | / | 0.080 | 0.330 | **0.999** | 0.933 | 0.782 | 0.404 | 0.320 |
| | Ultrasound - Blur | 0.001 | 0.000 | / | / | / | 0.386 | 0.330 | **0.967** | 0.958 | 0.790 | 0.411 | 0.636 |
| LiDAR | [Hide] Laser - Point Erase | / | / | 0.655 | 0.645 | 0.661 | 0.597 | 0.638 | 0.653 | 0.676 | **0.683** | 0.611 | 0.665 |
| | [Create] Laser - Object Injection | / | / | 0.781 | 0.778 | 0.767 | 0.775 | **0.890** | 0.793 | 0.796 | 0.707 | 0.830 | 0.889 |
| | Laser - Arbitrary Point Injection | / | / | 0.893 | 0.873 | 0.894 | 0.784 | 0.892 | 0.890 | **0.910** | 0.884 | 0.875 | 0.888 |
| | Laser - Background Noise Injection | / | / | 0.814 | 0.855 | 0.742 | 0.516 | 0.898 | 0.854 | 0.922 | 0.729 | 0.839 | **0.959** |
| | EM - Point Interference | / | / | 0.979 | 0.987 | 0.981 | 0.960 | 0.985 | 0.971 | **0.994** | 0.993 | 1.001 | 0.993 |
| Mean Robustness on Camera Cor. ($mRb^C$) | | 0.359 | 0.312 | / | / | / | 0.511 | 0.630 | **0.988** | 0.977 | 0.844 | 0.657 | 0.707 |
| Mean Robustness on LiDAR Cor. ($mRb^L$) | | / | / | 0.825 | 0.827 | 0.809 | 0.726 | 0.861 | 0.824 | 0.850 | 0.799 | 0.831 | **0.879** |
| Mean Robustness on All Corruptions ($mRb$) | | 0.650 | 0.625 | 0.920 | 0.922 | 0.913 | 0.609 | 0.735 | 0.918 | **0.923** | 0.824 | 0.737 | 0.785 |

collisions or traffic jams as intended by the attacker. Past research has shown that single-modality models are notably susceptible to such targeted attacks, highlighting the unique and significant threats posed by targeted attacks that necessitate the attention of security researchers. Consequently, evaluating whether attackers can control the outputs of multi-modal models in a straightforward manner, similar to their control over single-modality models, serves as one of the key aspects in measuring model robustness.

We use the parameter *ASR* to evaluate targeted attack robustness. The results are shown in Table 1. We take one camera-based detector ImVoxelNet [62] and one LiDAR-based detector PointPillar [44] as the baseline. It can be observed that the 4 targeted attacks can achieve a high attack success rate against the single-modality detectors. In contrast, the ASR of the attacks on the fusion models varies significantly. Overall, **hiding is easier than creating**. Examining each type of attack, the LiDAR-Hide attack can successfully compromise all models, as the method of erasing point clouds can almost entirely eliminate an object's 3D information. This action consequently prevents the successful regression of the 3D bounding box. Continuing with this line of reasoning, Camera-create, which does not provide 3D information about the object, should logically be unable to succeed. This holds true for the majority of models. However, we were surprised to find that the Camera-Create attack successfully generated spoofed objects in F-pointnet. To understand why the Camera-Create attack succeeds, we visualized the detection results as shown in Fig. 8. We found that the created objects in F-pointnet are instances where the ground is mis-detected as a car. We suppose that after the filtering mechanism in F-pointnet, the

ground point cloud obtained features closely resembling those of a car roof. Moreover, the success rates of Camera-Hide attacks and LiDAR-Create attacks are inversely related. Models easily compromised by Camera-Hide attacks are less susceptible to LiDAR-Create attacks and vice versa. This indicates that existing fusion models often rely more on one sensor source (majority). The late fusion in CLOCs treats detection results from both modalities equally, eliminating this bias. However, due to the structural characteristics of CLOCs, it tends to prune rather than create new discoveries, which can be hazardous in real autonomous driving scenarios.

> **Observation 1** (RQ1): For camera attacks, all fusion models (7/7) can reduce the attack success rate. However, no fusion model (0/7) can defend against the LiDAR-Hide attack, and only some models (4/7) can attenuate the LiDAR-Create attacks. The black-box targeted attacks, originally designed against single-modality detectors, still retain the potential capability to compromise fusion models. However, selecting the right fusion model can effectively mitigate the impact of such attacks.

*4.4.2 RQ1.2 Single Source Robustness:* Single Source Robustness refers to the average robustness of a model when facing attacks on a single sensor (such as a camera or LiDAR). Since single-modality models are only exposed to attacks targeting the sensor they use, single-source robustness allows for a comparison of the robustness between MSF and single-modality models when confronting the same attacks.

We utilize the parameters $mRb^C$ and $mRb^L$ to represent the model's mean robustness under camera or LiDAR corruption, respectively. The results are shown in Table 2, We found that the $mRb^C$ of all MSF models surpasses that of camera-based models. This improvement can be attributed in part to the fusion of point clouds, which effectively enhances robustness. Another contributing factor is the generally inferior performance of existing open-source camera-based models, leading to their lower robustness. In contrast, only 4 out of the 7 MSF models in our experiments showed a superior $mRb^L$ to LiDAR-based models. This suggests that fusion doesn't necessarily guarantee enhanced robustness, and selecting the right fusion method requires additional effort.

> **Observation 2** (RQ1): When considering single source attacks: Compared to the camera-based model, all fusion models (7/7) can enhance the robustness against camera attacks. Compared to LiDAR-based models, most fusion models (5/7 in this paper) can increase robustness against LiDAR attacks.

*4.4.3 RQ1.3 Overall Robustness:* Overall robustness refers to the model's mean robustness under all corruptions in this benchmark. Given the increased diversity of sensors in the MSF-based models, they are exposed to a greater number of potential attack vectors. This aspect is pivotal and cannot be ignored when evaluating robustness and security.

We use the parameter $mRb$ to evaluate overall robustness. For Camera-based models, we set their robustness to all LiDAR corruptions as 1, and vice versa for the LiDAR-based models. From Table 2, we observe that the majority of MSF models have greater $mRb$ compared to camera-based models. This discrepancy may be attributed to the subpar performance of existing open-source camera-based models. The best $mRb$ is exhibited by the LiDAR-based model. Meanwhile, the SOTA MSF models, VirConv-L and VirConv-T, also demonstrate commendable $mRb$. Overall, MSF-based models have a lower mRb compared to LiDAR-based models.

> **Observation 3** (RQ1): Since MSF-based models are exposed to sensor attacks from both sensors, the majority (6/7) of existing fusion models do not enhance overall robustness.

---

**Answer to RQ1**

Considering *targeted attack robustness*, *single source robustness*, and *overall robustness*, most MSF-based models show enhanced robustness compared to camera-based models rather than LiDAR-based models. However, state-of-the-art fusion models, such as VirConv-L and VirConv-T, are expected to enhance robustness across all aspects, showcasing the potential of MSF in improving robustness.

---

## 4.5 RQ2. How does the architecture of the model influence robustness?

To answer RQ2, we compare MSF-based models with each other, aiming to evaluate which fusion design is more robust. We approach this comparison from two perspectives: fusion sequence and fusion data representation.

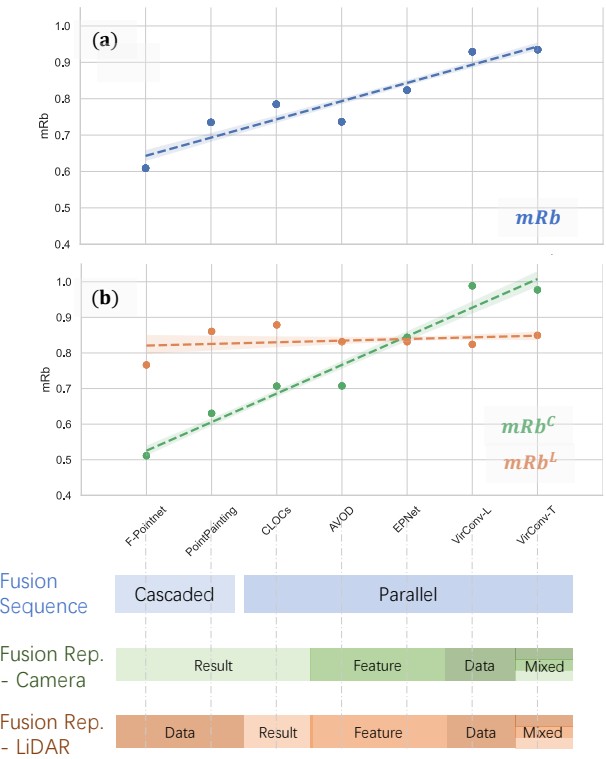

**Figure 6: The mean robustness under (a) all corruptions and (b) camera and LiDAR corruptions.** The models on the X-axis are ranked based on two criteria. The first criterion is the fusion sequence, in the order of [cascaded, parallel]. Within the same fusion sequence, the information entropy of fusion representation serves as the second criterion, in the order of [result, feature, data].

*4.5.1 Fusion Sequence.* According to the results in Table 2, the robustness of a model varies greatly with different architectures. To discern the relationship between fusion sequence and robustness, we categorize the models into cascaded fusion and parallel fusion. As the $mRb$ under all corruptions shown in Fig. 6, we observed that the robustness of cascaded fusion is generally lower than that of parallel fusion. We suppose this is due to the cascade effect, where errors caused by corruption in a single sensor propagate throughout the detection pipeline, subsequently reducing overall robustness. In contrast, parallel fusion allows data from the two sensors to reinforce each other, thereby enhancing robustness.

> **Observation 4**(RQ2) From the perspective of the fusion sequence, parallel fusion exhibits better robustness than cascaded fusion.

*4.5.2 Fusion Representation.* From the input to the output of the detection pipeline, data representation transitions from original *data* to *feature* and then to *results*. We adopt information entropy, denoted as $\mathcal{H}_X$, to intuitively quantify the information content of data $X$. In neural networks, basic operations such as convolution, activation functions, ROI pooling, NMS, and FC can lead to information loss [28]. Thus, we can intuitively derive the following relationship:

$$\mathcal{H}_{data} > \mathcal{H}_{feature} > \mathcal{H}_{result} \tag{3}$$

Additionally, We use $\mathcal{H}_{FR}(M)$ to represent the information entropy contained in the Fused Representation of model $M$.

First, let's consider the cascaded models: F-Pointnet and Point-Painting. These two models both use 2D results generated from the image to fuse with the original point cloud. While F-Pointnet reduces the information entropy of the point cloud by filtering the point cloud using 2D detection results. we have:

$$\mathcal{H}_{FR}(Pointpainting) > \mathcal{H}_{FR}(F-pointnet) \quad (4)$$

Second, let's consider the parallel fusion models VirConv-T, VirConv-L, EPNet, AVOD, and CLOCs. The fused representations of the five models are shown in Fig. 6. It's important to note that the LiDAR input of AVOD is a bird's-eye view (BEV). Clearly, the information entropy of BEV is less than that of the original point cloud. Thus, we can determine that $\mathcal{H}_{FR}(AVOD)$ is less informative than $\mathcal{H}_{FR}(EPNet)$, but we cannot compare $\mathcal{H}_{FR}(AVOD)$ and $\mathcal{H}_{FR}(CLOCs)$. Therefore, we have:

$$\mathcal{H}_{FR}(VirConv-T) > \mathcal{H}_{FR}(VirConv-L) >$$
$$\mathcal{H}_{FR}(EPNet) > \mathcal{H}_{FR}(AVOD), \mathcal{H}_{FR}(CLOCs) \quad (5)$$

As shown in Fig. 6(a), the overall robustness of the models also follows the information entropy relationship in Equ. 4 and Equ. 5. This confirms the *Observation 5*.

> **Observation 5** (RQ2): In general, given the same fusion sequence, the more comprehensive the information contained in the fused representation, the stronger the robustness. The comprehensiveness of information is ranked as data > feature > results.

Further analysis, as shown in Fig. 6(b), reveals that different fusion representations primarily influence the $mRb^C$. Moreover, the greater the information entropy ($\mathcal{H}_{FR}$), the stronger the $mRb^C$. However, the variation of $\mathcal{H}_{FR}$ has very few impacts on the $mRb^L$. This is because those MSF models are based on point cloud-based 3D object detectors and incorporate image information into various stages of the detection pipeline.

> **Answer to RQ2**
>
> Overall, different fusion sequences and fusion representations influence robustness with the following characteristics:
> 1) Parallel fusion exhibits better robustness than cascaded fusion. 2) The more comprehensive the information contained in the fused representation, the greater the robustness. The comprehensiveness of information is ranked as data > feature > results. 3) Different fusion architectures primarily influence the robustness to camera corruption.

## 5 ROBUSTNESS IMPROVEMENT

Based on the answers of RQ1 and RQ2, we provide insights for enhancing robustness. A Fusion architecture exhibiting the following characteristics can enhance robustness against physical sensor attacks: 1) Independence, 2) Parallel Fusion, and 3) Data Fusion.

**Independence** refers to the ability of each modality to achieve the final 3D object detection independently of the others. This allows one modality to potentially complete the final task even when subjected to data erasure attacks, such as the camera-hide and lidar-create discussed in this paper.

**Modality Equality** refers to incorporating sensor data equitably into the detection model during fusion, rather than designating one sensor as primary and another as auxiliary. Empirical evidence has shown that if bias is introduced towards one modality during fusion, the model becomes more vulnerable to attacks targeting the primary sensor. Moreover, the auxiliary sensor, having insufficient weight, faces challenges in effectively correcting the outcomes.

**Data Fusion** suggests that integration should occur at the raw data level, rather than at the feature or result levels. This is because raw data retains more comprehensive information, and experimental findings have affirmed the increased robustness of data fusion compared to other fusion methods. However, when fusing LiDAR and camera data, the process encounters challenges due to the heterogeneous nature of point clouds and images, which impedes direct data fusion.

## 6 RELATED WORK

Understanding and analyzing the robustness of fusion-based perception has been broadly studied with digital data corruption (e.g. occlusion [38, 57], noise [38, 55] or downsampling [55]) and worst-case adversarial perturbation [55, 75, 86]. While these works bring intriguing results in most cases, they share two limitations. First, the corrupted data they adopted is purely digital, not reflecting the challenges fusion systems might encounter in the real world. Second, the number of fusion models they tested is limited (fewer than 3) and the performance of models is not state-of-the-art. This could potentially undermine the validity of the empirical conclusions drawn, which may lead to contradictory conclusions in different works. For instance, [57] said *"the later the sensor data is fused, the greater the detection rate of object detectors is"*, while [75] said *"early fusion is more robust than late fusion"*.

## 7 CONCLUSION

In this paper, we introduce, to our knowledge, the first comprehensive benchmarks for MSF-based Models under physical sensor attacks by introducing a new dataset including 11 types of sensor attacks. We designed and conducted a rigorous SLR and attack capability quantification to ensure the comprehensiveness and physical feasibility of Kitti-Spoof as much as possible. Based on evaluating 542,736 frames on 7 MSF-based models and 5 single-modality models, we answer two open research questions: *RQ1) Does fusion enhance robustness?* We find most fusion models reduced overall robustness when considering attacks from both sensors. This challenges the consistent understanding of previous research. *RQ2) How does the architecture of the model influence robustness?* We adopted a novel paradigm to categorize models and introduced the concept of information entropy, which surprisingly revealed the relationship between model architecture and robustness. Finally, we provided some insights for enhancing robustness. We hope that our benchmark can aid in improving the performance of MSF-based models. The study can serve as a reference for researchers concerned with the security of MSF-based models in safety-critical applications.

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

# A ATTACK CAPABLITY QUANTIFICATION

## A.1 Image Corruption

**[Camera - Hide]Laser-Saturation:** Previous papers have conducted the hiding attack on a camera using LED and laser, and the experimental results showed that laser can easily blind the camera even damage it compared to LED. In the previous experiments, only the red laser ( 650 nm) was used. In order to make the experimental results more complete, in this paper, we use more wavelengths of lasers for the experiments, and we also tried high lumen beams. We discovered that at the same power, green lasers( 550 nm) can cause more severe overexposure effects than lasers of other wavelength. This may be due to the higher proportion of green pixels in CMOS sensors.

Table 3: The Attack Capability of The Transduction Attack Corruptions

| ID | Corruptions | Attack Capability | | | Source Paper |
| --- | --- | --- | --- | --- | --- |
| | | Corruption Patterns | $\ell_0$ :Scope | $\ell_2$ :Pixel / Point Manipulation Quantification | |
| 1 | [Hide] Laser - Saturation | Global Exposure | All Piexels | Value addition on {R,G,B} channels according to quantum efficiency | [56, 83] |
| 2 | [Create] Light- Projection | Specified pattern | Specified location | Value superposition of projected pixels and original pixels | [51],[53],[48] [21],[77] |
| 3 | Laser - Color Strip Injection | color strip | Specific rows of the image. | Value addition on {R,G,B} channel according to quantum efficiency | [65, 84] |
| 4 | EM - Color Strip Loss | Uniform color strip | Specific rows of the image. | Filter array mismatch: G→R/B, R/B→G | [35] |
| 5 | EM - Truncation | Content Stitching | Specific rows of the image. | The image is stitched with the previous frame or next frame | [35] |
| 6 | Ultrasound - Blur | Linear Blur | All Piexels | Value superposition of a series of translated pixels | [34, 89] |
| 7 | [Hide] Laser - Point Erase | Point Erase | 30° azimuth | The original points are erased | [12, 36] |
| 8 | [Create] Laser - Object Injection | Specified Object | 20° azimuth | Fake points with random distance noise of 0.05 meters. | [19, 36, 64] |
| 9 | Laser - Arbitrary Point Injection | adversarial point cloud | 30° azimuth | Fake adversarial points with raondom distance noise of 1 meters | [14, 56, 67] [29, 69] |
| 10 | Laser - Background Noise Injection | unfirom noise | 30° azimuth | Fake random points within a distance of 100 meters | [67] |
| 11 | EM - Point Interference | sinusoidal noise | All Points | Sinusoidal noise wintin 0.05m added to the original points | [11] |

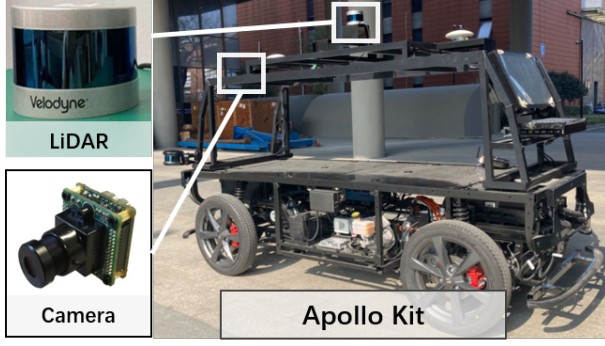

Figure 7: Fusion Testbed.

**[Create]Light - Projection:** We conduct tests with the two methods: projecting into the environment and directly projecting into the camera. However, achieving a successful direct projection into the camera proved challenging, as it necessitates precise optical focusing of the projector and high-precision aiming between the attack signal and the camera's photosensitive components. Through testing, we finally choose to implement the attack by projecting patterns into the environment. We find this approach convenient for launching *create* attacks and can effectively deceiving state-of-the-art 2D object detection models. However, the projection attacks can be notably challenging to execute successfully under strong lighting conditions.

**Laser - Color Strip Injection:** The authors of "Rolling Color" extensively discuss the impact of pulsed lasers on images in their paper and provide a modeling method for Laser Attack. In this paper, we adopt their approach for designing corruption.

**EM - Strip Loss and EM - Truncation:** The GlitcHike demonstrates that attackers can utilize EMI to induce a color strip in images due to errors in the optical filters, such as the incorrect use of blue-green (B/G) and green-red (G/R) filters. As a result, the injected strip visually appears as a uniform shade of purple (distinguishing it from the uneven strip in *Color Strip Injection*). The Glitchhike paper provides evidence that attackers can control the position, width, and number of purple strips. Similarly, for truncation, attackers can adjust the signal to control the position of truncation. We have also confirmed this in testbed-based testing. Therefore, we follow the attack capability outlined in the paper for designing corruption.

**Ultrasound - Blur:** Based on the three types of pixel motions along different Degrees of Freedom (DOFs), the authors [34] categorize the blur patterns into linear blur, radial blur, and rotational blur. Through the physical experiments in our testbed, we have observed that linear blur is the most easily induced type of blur. Therefore, we adopt linear blur to design corruption.

## A.2 LiDAR Corruption

**[Hide]Laser - Point Erase:** In the paper [67] by Shin et al., they utilized an 800mW continuous laser to hide point clouds of a $41*42cm^2$ metal plate, but they did not quantify the specific attack capabilities. In our testbed experiments, we conducted experiments using 905nm lasers with power outputs of 200mW, 600mW, 1000mW, and 2000mW, respectively. We found that as the power increases,

**Table 4: MSF-based 3D object detection model in our benchmark**

| Model | Fusion Stage | Fusion Architecture | Fusion Representation | | Fused Operator |
|-------|--------------|---------------------|----------------------|---|----------------|
| | | | Camera Rep. | LiDAR Rep. | |
| F-Pointnet | Early | Cascaded | frustum | point cloud | region selection |
| PointPainting | Early | Cascaded | 2D segmentation | point cloud | point-wise enhancement |
| VirConv-L | Early | Parallel | virtual points | point cloud | data concatenate |
| VirConv-T | Mixed | Parallel | virtual points | point cloud | data concatenate & ROI concatenation & box voting |
| EPnet | Middle | Parallel | image features | point features | point-wise attention |
| AVOD | Middle | Parallel | image features | BEV features | concatenation & MLP |
| Clocs | Late | Parallel | 2D boxes | 3D boxes | box consistency |

the effective range of removal also increases. Using a 2000mW laser, we were able to erase point clouds within approximately a $6° * 6°$ area. In the studies by Jin et al. [36] and Cao et al. [12], they conducted a detailed evaluation, demonstrating that attackers can remove target point clouds over a horizontal angle of more than 30° [36] and 40° [12], respectively. We have also confirmed this on our testbed. Considering the overall attack effectiveness and cost, we have decided to draw inspiration from the latter attack method for designing our corruption approach.

**[Create]Laser - Object Injection:** The authors of PLA-LiDAR claim the capability to control up to 4000 points within 30°, a number sufficient for injecting point clouds of objects such as cars and pedestrians. In addition to the number points, to enhance the physical realizability, the position and shape control precision of points are crucial metrics need to be considered. Position precision refers to the attacker's ability to precisely control the overall position of the injected point clouds. Shape precision refers to the ability to maintain the injected points into specific shapes. Based on calculations of continuous data for 7 seconds (70 frames), we observed that the position and shape precision follows random uniform distribution. When we set the mean value is 0, the standard deviation of position precision is 38.2cm and standard deviation of the shape precision is 5.3cm. We take these two errors into consideration when designing the corruptions.

**Laser - Arbitray Point Injection:** In these previous works, the latest research claims the capability to inject up to 200 points. However, inspired by PLA-LiDAR, we believe that injecting thousands of points is feasible. Therefore, when designing corruption, we set the number of points to be the same as in *Create-Object Injection* and assign each point a mean error ranging from 0 to 1 meter to reflect its randomness.

**Laser - Background Noise Injection:** In the paper [67], Shin et al. claim the ability to inject noise within a 20° horizontal angle. However, this information alone is insufficient for designing corruption. Therefore, we conduct further experiments on our testbed using 5mW and 30mW continuous 905nm lasers at a distance of 7 meters. We find that it is feasible to inject noise within approximately a 30° horizontal angle range, and the noise is uniformly distributed within a range of 0 to 150 meters

**EM - Point Interference:** In the paper [11], Bhupathiraju et al. utilized EM signals at frequencies of 960.9MHz and 977.4MHz to induce sinusoidal and random patterns in the LiDAR's point cloud. Meanwhile, they achieved an average displacement of approximately 4cm under a 25dB EMI power. We follow the attack capability outlined in the paper for designing corruption.

## B MODEL ARCHITECTURE ANALYSIS

In this section, we evaluated the ASR of the targeted attack against the fusion model, We take two single-modality detectors YOLO v8 [9] and PointPillar [44], as the baseline. The results are shown in Table. 1. For LiDAR-Hide attack, the target objects are those with all eight vertices of the ground-truth 3D box within the attack range. It can be observed that the 4 targeted attacks can achieve high attack success rate against the single-modality detectors. In contrast, the impact of the attacks on the fusion models varies significantly depending on the model structure. Overall, hiding is easier than creating. Among these, the LiDAR-Hide attack can successfully compromise all models, as the method of erasing point clouds can almost entirely eliminate an object's 3D information. This action consequently prevents the successful regression of the 3D bounding box. Continuing along this line of reasoning, Camera-create, which does not provide 3D information about the object, should logically be unable to succeed, and indeed, this holds true for the majority of models. But we found that Camera-create can successfully attack F-pointnet, and we provide an explanation latter.

We provide further explanations for the results of other two attacks, analyzing them on a model-by-model basis.

**F-Pointnet.** Due to the cascaded fusion structure of F-pointnet, if an object is not detected in the image, then the object's point cloud will be filtered out, leading to a high success rate for the Camera-Hide attack. It is important to note that, strictly speaking, the Camera-Create attack cannot succeed in all models since merely modifying an image does not provide 3D information. However, we were surprised to find that the Camera-Create attack successfully generated spoofed objects in F-pointnet. To understand why Camera-Create attack succeeds, Upon manually analyzing the results, we visualize a typical 3D detection results in . we discovered that the spoofed objects created in F-pointnet were instances where

Table 5: The Performance(AP) of 11 Detectors on Kitti and Kitti-Spoof.

| Target Sensor | Corruption | Difficulties | Camera-basd | | LiDAR-based Model | | | Fusion Model | | | | | | |
|---|---|---|---|---|---|---|---|---|---|---|---|---|---|---|
| | | | ImVoxelNet | SMOKE | Second | PointPillar | 3DSSD | F-PointNet | PointPainting | VirConv_L | VirConv_T | EPNet | AVOD | CLOCs |
| | AP on Clean Data | Easy | 22.320 | 16.936 | 88.356 | 87.813 | 88.892 | 72.052 | 86.360 | 89.944 | **90.089** | 88.922 | 82.724 | 87.391 |
| | | Moderate | 17.271 | 13.848 | 78.214 | 77.616 | 78.434 | 65.150 | 76.725 | 86.818 | **87.731** | 78.831 | 73.717 | 76.855 |
| | | Hard | 15.164 | 11.902 | 76.017 | 75.931 | 77.326 | 53.162 | 75.091 | 85.912 | **86.508** | 78.400 | 67.650 | 75.113 |
| Camera | [Hide] Laser - Saturation | Easy | 4.545 | 1.299 | / | / | / | 14.301 | 37.906 | 89.962 | **90.051** | 77.574 | 52.725 | 28.313 |
| | | Moderate | 9.091 | 0.826 | / | / | / | 14.530 | 31.107 | **86.477** | 86.302 | 60.320 | 42.723 | 23.295 |
| | | Hard | 9.091 | 0.826 | / | / | / | 14.206 | 26.742 | **85.955** | 84.956 | 60.097 | 37.312 | 23.751 |
| | [Create] Light- Projection | Easy | 14.621 | 13.620 | / | / | / | 24.927 | 84.645 | 89.812 | **90.040** | 88.862 | 82.350 | 85.143 |
| | | Moderate | 11.167 | 11.564 | / | / | / | 33.271 | 75.520 | 86.717 | **87.847** | 78.729 | 73.339 | 75.874 |
| | | Hard | 10.814 | 11.178 | / | / | / | 30.738 | 71.581 | 85.882 | **86.572** | 78.304 | 67.269 | 74.490 |
| | Laser - Color Strip Injection | Easy | 10.292 | 3.422 | / | / | / | 69.816 | 75.040 | **89.958** | 89.900 | 77.486 | 67.194 | 86.963 |
| | | Moderate | 9.091 | 3.314 | / | / | / | 60.071 | 62.226 | 86.498 | **87.311** | 59.561 | 54.041 | 76.209 |
| | | Hard | 9.091 | 1.946 | / | / | / | 53.235 | 60.821 | **86.039** | 85.208 | 59.258 | 47.384 | 74.544 |
| | EM - Color Strip | Easy | 10.472 | 11.167 | / | / | / | 66.343 | 80.114 | **89.810** | 89.480 | 86.649 | 68.751 | 86.828 |
| | | Moderate | 9.878 | 10.580 | / | / | / | 60.418 | 70.474 | 85.466 | **86.646** | 68.652 | 54.432 | 76.146 |
| | | Hard | 9.718 | 10.213 | / | / | / | 53.592 | 67.465 | 78.766 | **84.186** | 63.966 | 53.873 | 74.547 |
| | EM - Truncation | Easy | 0.006 | 0.000 | / | / | / | 2.910 | 32.172 | **89.959** | 89.289 | 76.963 | 37.819 | 28.419 |
| | | Moderate | 0.005 | 0.000 | / | / | / | 3.195 | 23.559 | **86.475** | 78.761 | 57.745 | 27.065 | 23.871 |
| | | Hard | 0.005 | 0.000 | / | / | / | 9.091 | 22.861 | **86.009** | 78.501 | 57.760 | 25.627 | 24.385 |
| | Ultrasound - Blur | Easy | 0.001 | 0.000 | / | / | / | 31.804 | 32.172 | **89.754** | 89.544 | 77.168 | 38.089 | 57.379 |
| | | Moderate | 0.000 | 0.000 | / | / | / | 22.966 | 23.559 | **85.557** | 84.572 | 58.867 | 27.346 | 46.800 |
| | | Hard | 0.000 | 0.000 | / | / | / | 18.648 | 22.861 | 78.820 | **78.980** | 58.407 | 26.625 | 48.013 |
| LiDAR | [Hide] Laser - Point Erase | Easy | / | / | 60.232 | 59.041 | 60.766 | 44.483 | 58.271 | 67.016 | **69.093** | 64.308 | 53.672 | 60.071 |
| | | Moderate | / | / | 50.404 | 49.664 | 51.147 | 35.517 | 48.028 | 52.500 | **56.770** | 52.132 | 41.929 | 50.407 |
| | | Hard | / | / | 48.352 | 46.978 | 49.74 | 33.580 | 45.599 | 51.973 | **52.945** | 51.703 | 41.418 | 48.608 |
| | [Create] Laser - Object Injection | Easy | / | / | 72.864 | 70.113 | 69.573 | 60.669 | 77.854 | 73.615 | 71.1231 | 62.615 | 68.504 | **78.153** |
| | | Moderate | / | / | 59.509 | 60.241 | 59.707 | 55.175 | 67.992 | 65.325 | 64.26643 | 55.957 | 61.645 | **68.088** |
| | | Hard | / | / | 57.208 | 57.363 | 58.297 | 44.318 | 66.220 | 58.969 | 61.83261 | 55.405 | 55.928 | **66.452** |
| | Laser - Arbitrary Point Injection | Easy | / | / | 79.683 | 77.666 | 80.993 | 55.523 | 78.218 | 86.027 | **84.324** | 78.978 | 73.174 | 78.240 |
| | | Moderate | / | / | 69.56 | 68.148 | 69.672 | 50.266 | 68.073 | 77.642 | **78.785** | 69.708 | 64.542 | 68.074 |
| | | Hard | / | / | 67.29 | 64.937 | 68.125 | 43.468 | 66.262 | 70.138 | **77.327** | 68.988 | 58.335 | 66.334 |
| | Laser - Background Noise Injection | Easy | / | / | 87.191 | 76.528 | 70.545 | 38.302 | 79.268 | 86.187 | **87.955** | 65.888 | 72.260 | 86.640 |
| | | Moderate | / | / | 72.592 | 66.831 | 57.392 | 32.461 | 69.196 | 69.528 | **78.301** | 58.486 | 58.327 | 75.157 |
| | | Hard | / | / | 67.689 | 62.937 | 53.649 | 27.486 | 65.305 | 68.551 | **77.421** | 55.034 | 57.518 | 67.784 |
| | EM - Point Interference | Easy | / | / | 87.646 | 87.369 | 88.583 | 69.554 | 84.795 | 89.853 | **89.966** | 88.506 | 82.924 | 87.023 |
| | | Moderate | / | / | 77.023 | 77.068 | 77.635 | 60.162 | 75.953 | 86.006 | **87.768** | 78.273 | 73.842 | 76.467 |
| | | Hard | / | / | 72.9 | 73.719 | 73.773 | 52.961 | 73.806 | 79.131 | **84.929** | 77.689 | 67.624 | 74.228 |

the ground was misdetected as a car (Fig. 8). We suppose that after the filtering mechanism in F-pointnet, the ground point cloud obtained features closely resembling those of a car.

**Pointpainting.** Camera-Hide and LiDAR-Create can achieve 47.41% and 95.93% ASRs, respectively, on Pointpainting, seperately. The PointPainting architecture consists of three main stages. (1) an image based semantic segmentation network which computes the pixel wise segmentation scores. (2) a fusion stage that LiDAR points are painted with sem. seg. scores. (3) a LiDAR based 3D detection network. According to the taxonomy in this paper, it's an early fusion. The image segmantation scores are appended to the LiDAR point to create the painted point. The paited point can be consumed by any LiDAR network that learns an encoder, since PointPainting just changes the input dimension of the LiDAR points. In this benchmark, we used the Pointpillar [44] and decorate the point cloud with the sem. seg. scores for 4 classes in Kitti. This changes the point cloud dimension from $9 \rightarrow 13$. The first 9-dimensions point cloud data and subsequent 4-dimensions image data are fed into

(13,64) encoder for 3D object detection. Therefore, both the image's classification information and the raw point cloud information have the potential to influence the final detection results. Furthermore, based on the results, it can be observed that the fusion architecture of point painting exhibits greater robustness against camera attacks compared to LiDAR attacks.

**EPNet.** EPNet is very robust against Camera-Hide (ASR 7.78%) while is vulnerable to LiDAR-Create(ASR 100%). EPNet consists of a two stream backbone network, which is composed of a geometric stream and an image stream. The two streams produce the point features and semantic image features, respectively. In imgae stream, EPNet adopts four cascaded $3 * 3$ convolutional blocks to extract image semantic features in different scales, denoted as $F_i(i = 1, 2, 3, 4)$. The geometric stream comprises four paired set abstraction $S_i(i = 1, 2, 3, 4)$ and feature propogation layers $P_i(i = 1, 2, 3, 4)$ [59] for feature extraction. The point features $S_i$ are combined with the image features $F_i$ with the aid of LI-Fusion

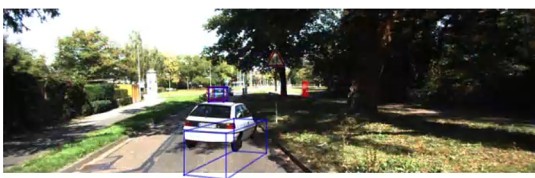

(a) Image

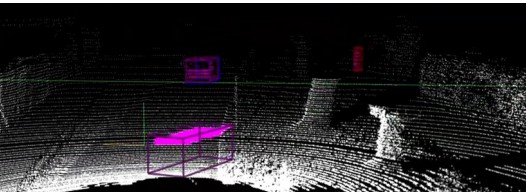

(b) Point Cloud.

**Figure 8: The detection results of F-Pointnet against [Create]Light-Projection corruption. The ground point clouds are misdetected as a "car", indicating the success of the create attack.**

module. Besides, the point feature $P_4$ is enriched by the image feature. Overall, the LI-Fusion module allows point features and image features to be deeply fused in the backbone network. However, as the primary role of the image is to enhance the point cloud, point cloud features still play a predominant role throughout the pipeline. Thus, it is easier for LiDAR-based corruptions to compromise EPNet compared to camera-based corruptions.

**AVOD.** AVOD is robust against Camera-Hide (ASR 20.48%) while is vulnerable to LiDAR-Create(ASR 100%). In AVOD, the image and bird-eye-view(BEV) point feature maps, which are generated by feature extractors, are fused in region proposal network(RPN) Both feature maps are then used by the RPN to generate non-oriented region proposals, which are passed to the detection network for dimension refinement, orientation estimation, and category classification. We can observe that the fusion architecture of AVOD puts the features of images and point clouds on an equal footing, which distinguishes it from EPNet. In EPNet, image features are used to aid in enhancing point cloud features. Although the AVOD architecture is intended to be equal for camera and LiDAR features, it is apparent that after training, the model is biased towards relying on the LiDAR features. This is evidenced by the attack results, where mean ASRs of camera attack vs LiDAR attack are 10.24% vs 100%.

**CLOCs.** Camera-Hide (ASR 88.51%) is able to successfully compromise CLOCs, whereas LiDAR-Create(ASR 0%) isn't. CLOCs is a late fusion approach that merges camera and LiDAR detection candidates before applying Non-Maximum Suppression (NMS). CLOCs employs significantly reduced thresholds for each sensor to optimize their recall rate. If the 2D and 3D bounding boxes have a large enough IoU in the image plane, then their information will be combined into a single tensor for subsequent processing. However, 3D(2D) bounding boxes for which no matching 2D(3D) bounding box can be found will be ignored. Overall, CLOCs, like most late fusions, tend to prune rather than create new discoveries, which explains why hide attacks are easy to succeed in while create attacks are difficult.

# C BACKGROUND

## C.1 MSF-based 3D Object Detection

Camera provides detailed texture information but is passive in dependence on suitable illumination. LiDAR provides accurate depth information but provides sparse observation at long range. Camera and LiDAR are considered to be two complementary sensor types for 3d object detection. Many endeavors have been made to fuse the image information from camera and the point cloud information from LiDAR for better 3D object detection. The state-of-the-art fusion methods mainly incorporate image information into different stages of the point cloud detection pipeline.

## C.2 Sensor Attack against Camera and LiDAR

There has been a lot of works analyzing and studying the vulnerabilities of vision sensors in autonomous driving, including research on sensor vulnerability and adversarial attacks based on deep learning. In order to gain a better understanding of the threats that autonomous driving perception faces in the physical world, we perform a literature review on physical spoofing attacks against camera-based and LiDAR-based object detection. As shown in Table. ,

The output of camera (i.e. picture) can be manipulated by light(laser), sticky, ultrasound and electromagnetic. The High-lumen LED and continuous laser will saturate the CMOS/CCD and cause the camera to be blind [56, 83]. Although the automatic exposure mechanism can weaken the high-intensity light to a certain extent, experiments have proved that this type of attack is still effective on cameras with automatic exposure [56]. Taking advantage of the feature that the projector can directly project high-lumen images, the desired patterns can be directly inject into the camera [51]or can be projected onto the ground [53] and target object [48] to achieve the spoofing effect. By exploiting the rolling shutter of CMOS sensors, radiometric striping distortions can be injected through the time-modulated high frequency light signal [46, 65], and then interference the target classification [65]. What's more, attackers manage to inject a color stripe overlapped on the traffic light in the image using laser, which can cause a red light to be recognized as a green light or vice versa [84]. In order to realize the adversarial attack in the physical world, there are many endeavours that use stickers to manipulate the pixel value of the image [22, 32, 40, 74, 79, 88, 92]. As a seemingly simple way to implement, this type of work often needs to consider the robustness of adversarial examples under various distances, angles, and illuminations [50]. The out-of-band signal injection have also been explored to manipulate pixel values. By emitting delicately designed acoustic signals, an adversary can control the output of an inertial sensor, which triggers unnecessary motion compensation and results in a blurredarbitrary image [34]. Using intentional electromagnetic interference (IEMI), an attacker could induce controlled glitch images of a camera at various positions, widths, and numbers [35].

The output of LiDAR (i.e. point cloud) can be manipulated by laser and 3D object. Similar to camera, saturation attack also applies to LiDAR sensors, and can be induced by high-power continuous laser [67]. After the iterative efforts of several papers [14, 56, 67], the latest literature [36] proves that a large number of controllable points can be injected into the mechanical (spinning) LiDAR

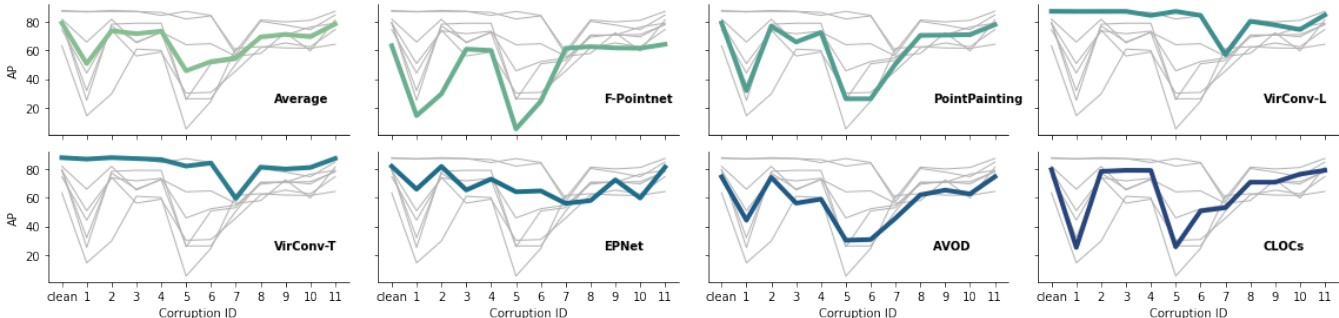

**Figure 9: The Average Precision (AP) of each model under clean datasets and various corruptions, a higher AP indicates better model performance.**

through carefully designed laser pulses, and physically validates the feasibility of hiding attack and creating attack. The way of using 3D objects to manipulate point clouds is also very popular. There are mainly two methods: 3D printing objects and placing arbitrary objects. 3D printing object can realize the adversarial point cloud of specific shape in the physical world, and make the attack difficult to be aware by human beings while spoofing the victim detection model [15, 71, 87]. Some studies have found that for the adversarial attack that aims to hide a target, the position of the adversarial points is more critical than the shape, thus the

adversarial effect can be realized by placing the arbitrary object at the specified position [90, 91].

For the LiDAR-Camera fusion Model, there are relatively few attacks implemented in the physical world. 3D printing object is employed to manipulate both point position changes in LiDAR point clouds and pixel value changes in camera images [13]. Although it has not been proved by systematic physical experiments, laser is considered as a promising way to spoofing LiDAR-camera fusion models [29]. Both of the above works emphasize maintaining the semantic consistency between LiDAR and Camera when designing attack vector.

