# OpenReview forum: "Unity is Strength? Benchmarking the Robustness of Fusion-based 3D Object Detection against Physical Sensor Attack"
_ACM.org/TheWebConf/2024/Conference — TheWebConf24 Oral_

### Official Review · Reviewer_mfmz · 2023-11-06

**Novelty:** 4
**Technical Quality:** 7

**Review:**

This work is a benchmarking work, firstly selects different kinds of physical sensor attacks, then create a dataset under each attack, evaluate the dataset upon 7 MSF-based models and 5 single-modality models. It provides some useful analysis regarding the robustness of using fusion models (e.g., fusion sequence, fusion representation)

Pros:
1. This work will release a large dataset, from the reviewer’s understanding, consisting of 11 different physical sensor attacks.

2. It comprehensively evaluates 7 latest MSF-models and 5 single-modality models.

3. It provides useful analysis when adopting MSF-models regarding e.g., fusion sequence, and fusion representation to potentially improve its robustness.

Cons:

1. The dataset link is not provided yet. Not sure on which condition and when the authors will release the dataset.

2. This is a benchmarking work; the novelty is limited or not the main contribution.

**Questions:**

Can the authors clarify whether the physical sensor attack can be realistically performed when the self-driving car is in a dynamic environment, e.g., fast-moving? It seems performing attacks in such a dynamic environment can be challenging.

The authors can justify why the first types of attacks in the first paragraph of Section 2.2 are out of the scope of their evaluation by linking to their 2.1 threat model.

The authors should state when they will make the dataset public, as this is the key contribution of this work.

**Reviewer Confidence:**

3: The reviewer is confident but not certain that the evaluation is correct

**Scope:**

3: The work is somewhat relevant to the Web and to the track, and is of narrow interest to a sub-community

---

### Official Review · Reviewer_xpJp · 2023-11-22

**Novelty:** 5
**Technical Quality:** 5

**Review:**

In this paper, Multi-Sensor Fusion (MSF) is proposed as a general strategy to improve the robustness of perceptual models. The robustness of the model in the face of different attacks is mainly studied, and advanced indicators based on IoU are introduced to evaluate the performance of the model. The experimental results show that the fusion model is more robust to some attacks than the single-mode model, but not all fusion methods can improve the robustness of the model, so it is necessary to choose the appropriate fusion method. Overall, this paper reveals the potential of model fusion to improve robustness through experiments and index analysis and provides ideas for further research.

Strengths
The paper introduces advanced IO-based metrics that can objectively evaluate the robustness of the model against different attacks.

The performance of different fusion methods is evaluated experimentally and a detailed analysis of the results is provided. This helps readers to understand the advantages and disadvantages of various methods and provides a reference for further research.

The paper also discusses the influence of different fusion methods on the robustness of the model through the analysis of information entropy. This provides researchers with ideas to design more robust fusion methods based on the characteristics of information entropy.




Weaknesses
The attack methods mentioned in the paper are relatively simple, and more complex attack scenarios may need to be considered to evaluate the robustness of the model more fully.

Although the fusion method proposed in this paper can improve the robustness of target detection, it may increase the complexity of computation and storage, and more work is needed to optimize the efficiency and resource utilization of the algorithm.

This paper analyzed few sensors, just LiDAR, and the camera。

**Questions:**

I thank the authors for their detailed review of multi-sensor fusion (MSF) as a general strategy for improving the robustness of perceptual models, as well as their evaluation of the robustness of multiple fusion approaches in the face of different attacks, and for their findings. It's an interesting job, a lot of work. They provide a comprehensive analysis of the robustness of multiple fusion methods in the face of different attacks and a comprehensive overview of multi-sensor fusion technology.
However, I still have some concerns outlined below:

In this paper, only the fusion analysis and evaluation of the two sensors of radar and camera are analyzed, which may cause different situations if other sensors are added.

The automatic driving system also has requirements for the real-time information obtained by sensors. Although the fusion method proposed in this paper can improve the robustness of target detection, it may increase the complexity of computation and storage, resulting in slow information transfer. More work is needed to optimize the efficiency and resource utilization of the algorithm.

The attacks in the Kitti-Spoof dataset were not comprehensive enough.

**Reviewer Confidence:**

2: The reviewer is willing to defend the evaluation, but it is likely that the reviewer did not understand parts of the paper

**Scope:**

3: The work is somewhat relevant to the Web and to the track, and is of narrow interest to a sub-community

---

### Official Review · Reviewer_sdHg · 2023-11-22

**Novelty:** 5
**Technical Quality:** 6

**Review:**

This paper focuses on the security and robustness of MSF in AD systems, particularly against physical sensor attacks.

Strength:

- Comprehensive Benchmarking
- Holistic Analysis of Sensor Attacks
- Novel Fusion Model Taxonomy

Weakness:

- Lack some details

**Questions:**

First, Implementing more robust systems as suggested by the paper, might incur additional costs or require significant changes in existing systems. The paper might not fully address these practical deployment challenges. Thus, I suggest the authors add some comparison of the cost on the systems.

In addition, the study primarily investigates LiDAR and camera sensors. Other sensor modalities, such as radar, ultrasonic, or thermal sensors, which are also used in autonomous vehicles, are not covered. This limits the understanding of the robustness of a complete sensor suite.

**Reviewer Confidence:**

3: The reviewer is confident but not certain that the evaluation is correct

**Scope:**

4: The work is relevant to the Web and to the track, and is of broad interest to the community

---

### Official Review · Reviewer_MGvD · 2023-11-28

**Novelty:** 2
**Technical Quality:** 3

**Review:**

The authors need to improve the grammar and sentence structure of the paper.

The paper aims to address the lack of benchmarking results of Multimodal sensory predictive models in autonomous vehicles and provide answers to the two following research questions: Does the multimodal approach improve the security and how does the architecture of the fusion model affect its robustness?

Both research questions are poorly defined and explained in the introduction section. Firstly, the author should be specific on what security they are referring to. By introducing factors such as bad weather and sensor failure, the problem became even more poorly defined. Secondly, when referring to the robustness of the multi-sensor fusion models, does the author refer to the training phase or the inference phase?

In the paper, the author lacks high level contribution and originality since MSF based papers already exist like the author mentioned. The author has not introduced any changes to existing models. The authors created a dataset in the field called KITTY-Spoof, but failed to explain how this dataset is relevant or comparable to real life data. Even their data from running existing models seems to suggest that existing MSP can already handle attacks or noisy data mentioned by the authors. The authors should also be more clear about how their work can provide a systematic approach to the field of research.

The conclusion behind the first research question seems trivial, since if a bias is added to the multimodal fusion model such that prediction from one sensor is preferred, the error rate will increase if the sensor is compromised.

**Questions:**

The author should answer the following question:

How can existing MSP be improved to better resist the aforementioned attacks?
How relevant is KITTY-Spoof to existing or real world performance?

**Reviewer Confidence:**

3: The reviewer is confident but not certain that the evaluation is correct

**Scope:**

1: The work is irrelevant to the Web

---

### Decision · Program_Chairs · 2024-01-22

**Decision:**

Accept (Oral)

**Comment:**

This paper focuses on the security of Multi-Sensor Fusion(MSF)-based 3D object detection in autonomous driving.The paper proposes a robustness benchmark for MSF-based 3D object detection under physical sensor attack. This benchmark encompasses five types of LiDAR attacks and six types of camera attacks. To further strengthen the benchmark, additional designs are incorporated to guarantee both comprehensiveness and physical feasibility.

 ## Strengths

 1. **New Benchmark.** The corruptions in this benchmark are induced by physical sensor attacks, which in this paper represent attacks that employ physical signals to manipulate sensor output. Unlike typical corruptions in previous robustness benchmarks-such as bad weather, digital noise, sensor failure, and object abnormalities-the corruptions in this work have a clear generation mechanism and can be intentionally induced by attackers. This benchmark fills a gap in previous robustness benchmark studies by specifically focusing on sensor attacks.

 2. **Comprehensiveness and Physical Feasibility of the Benchmark.**
 Comprehensiveness is achieved through an SLR(Scientific Literature Review)-based *corruption collection*, while physical feasibility of corruption is ensured by implementing *attack capability quantification*.

 3. **New Evaluation Methods and Insights.** The robustness evaluation in this paper revolves around two pivotal research questions: (1) Does fusion enhance security? and (2) how does the architecture of the fusion model influence robustness and provide insights. These led to novel observations and insights for enhancing the robustness of MSF-based models.

 ## Weaknesses

 1. The datasets can be released for further study